

# Facial expression recognition using visible and IR by early fusion of deep learning with attention mechanism

Muhammad Tahir Naseem[1], Chan-Su Lee[1], Tariq Shahzad[2],
Muhammad Adnan Khan[3,4,5], Adnan M. Abu-Mahfouz[2,6] and
Khmaies Ouahada[2]

[1] Department of Electronic Engineering, Yeungnam University, Gyeongsan, Republic of Korea
[2] Department of Electrical and Electronic Engineering Science, University of Johannesburg, Johannesburg, South Africa
[3] School of Computing, Skyline University College, Sharjah, United Arab Emirates
[4] Riphah School of Computing and Innovation, Riphah International University, Lahore, Pakistan
[5] Department of Software, Faculty of Artificial Intelligence and Software, Gachon University, Seongnam-si, Republic of Korea
[6] NextGen Enterprises and Institutions, Council for Scientific and Industrial Research, Pretoria, South Africa

Corresponding authors
Chan-Su Lee, chansu@ynu.ac.kr
Adnan M. Abu-Mahfouz,
aabumahfouz@csir.co.za

## ABSTRACT

Facial expression recognition (FER) has garnered significant attention due to advances in artificial intelligence, particularly in applications like driver monitoring, healthcare, and human-computer interaction, which benefit from deep learning techniques. The motivation of this research is to address the challenges of accurately recognizing emotions despite variations in expressions across emotions and similarities between different expressions. In this work, we propose an early fusion approach that combines features from visible and infrared modalities using publicly accessible VIRI and NVIE databases. Initially, we developed single-modality models for visible and infrared datasets by incorporating an attention mechanism into the ResNet-18 architecture. We then extended this to a multi-modal early fusion approach using the same modified ResNet-18 with attention, achieving superior accuracy through the combination of convolutional neural network (CNN) and transfer learning (TL). Our multi-modal approach attained 84.44% accuracy on the VIRI database and 85.20% on the natural visible and infrared facial expression (NVIE) database, outperforming previous methods. These results demonstrate that our single-modal and multi-modal approaches achieve state-of-the-art performance in FER.

## INTRODUCTION

Emotions, as neurophysiological changes, play a key role in human decision-making and social interaction. Research indicates that only 45% of communication is verbal, while 55% is conveyed through facial expressions (*Sun, Ge & Zhong, 2021*). The brain processes only key information in external data. Likewise, recognition models like CNNs should focus on essential features to enhance accuracy. Traditional facial emotion classification methods

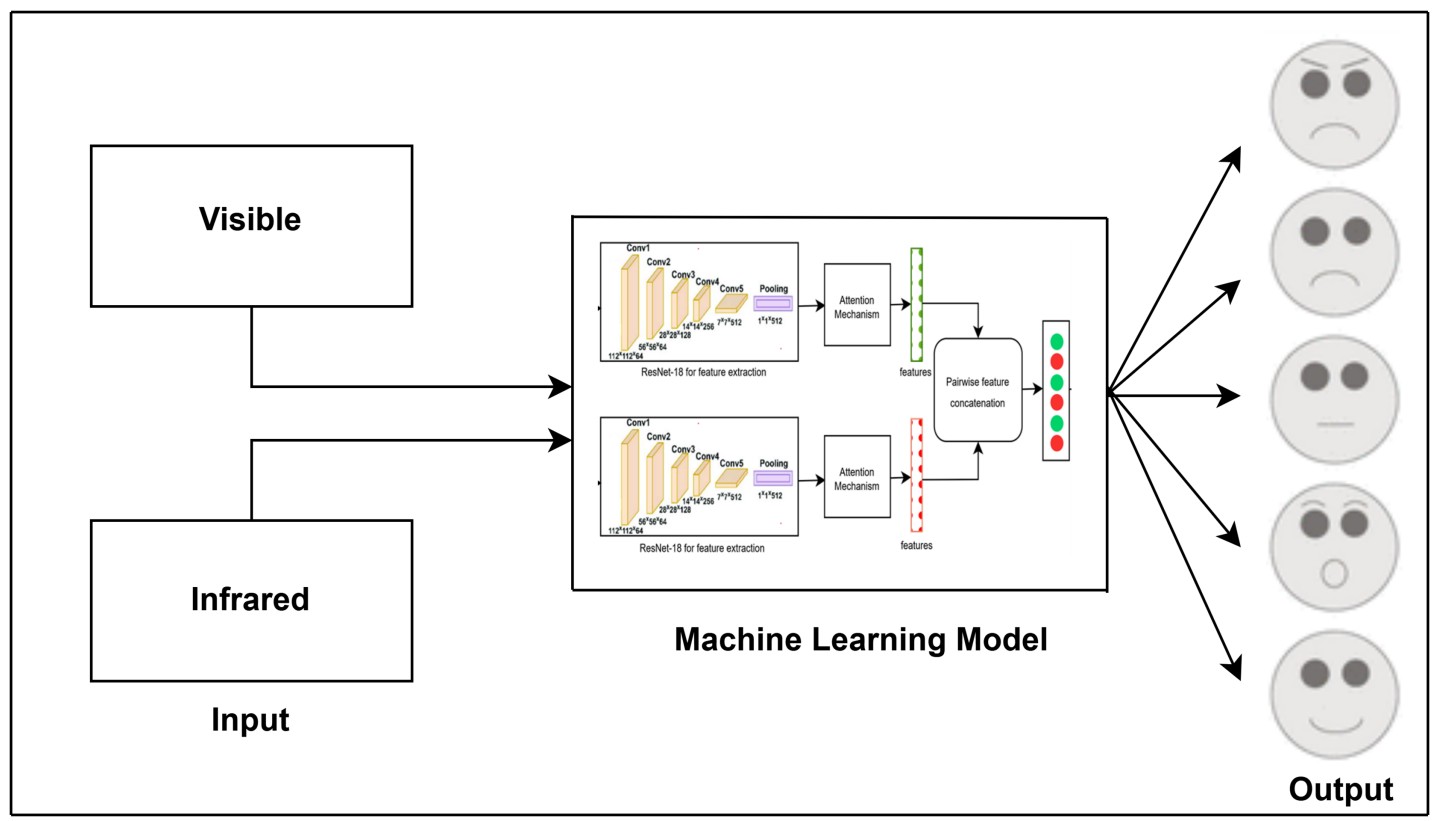

**Figure 1 Task flow diagram of the FER process from image input to emotion classification.**

often use algorithms like support vector machine (SVM) with handcrafted features such as local binary patterns (LBP) (*Li et al., 2020*).

The attention module has been widely used in several industries, such as saliency detection (*Zhao & Wu, 2019*), crowd counting (*Varior et al., 2019*), and facial expression identification (*Marrero Fernandez et al., 2019*). CNN-based techniques often use deep feature maps representing global image features. However, studies show a strong link between emotions and specific facial regions, though standard CNNs treat all areas equally (*Happy & Routray, 2014*). Figure 1 provides an overview of the facial expression recognition (FER) workflow: facial images (*e.g.*, visible, infrared) are captured (left), processed for feature extraction and classification (center), and produce recognized emotions such as anger, sadness, and happiness (right). This diagram illustrates the sequential steps of capturing, processing, and classifying emotions.

To identify five different facial expressions—angry, happy, neutral, sad, and surprised—available in the VIRI database and three different expressions—happy, disgust and fear available in the natural visible and infrared facial expression (NVIE) database, early and late fusion approaches were proposed (*Naseem, Lee & Kim, 2024*). The technique utilized different sets of visible, infrared, and multispectral dynamic imaging (MSX) images from the publicly available VIRI database and the NVIE database. In the early

fusion method, facial expressions (FEs) were recognized by concatenating features from different modalities. For the late fusion method, a weighted sum approach was used to combine the outputs of the modalities, with various weighting criteria on the basis of the accuracy of each individual model. The outcomes of the results showed that the suggested approach exceeded the results of earlier studies, achieving an accuracy of 83.33% with one-step training. By incorporating additional fine-tuning through three-step training, the accuracy improved to 84.44%. Further experiments included combining the models with another modality (MSX) available in the database, confirming that the additional modality, when integrated with visible and infrared modalities, could enhance the performance of fusion models. These findings highlight the effectiveness of multi-modal fusion techniques in improving the accuracy of facial expression recognition systems. The integration of multiple imaging modalities provides a more robust and reliable approach to identifying complex facial expressions.

This article aims to improve FER accuracy using an attention mechanism within single-modal and multi-modal fusion frameworks. Specifically, we (1) develop attention-enhanced models for visible and infrared modalities, (2) integrate these modalities *via* pair-wise feature concatenation, and (3) evaluate performance on VIRI and NVIE FER databases. We hypothesize that attention-based multi-modal fusion will outperform single-modal methods by capturing complementary features, enhancing accuracy and robustness in emotion classification.

The following are the contributions of our proposed work.

- We presented an early fusion approach that combines features from both visible and infrared modalities using pair-wise concatenation. This method leverages the strengths of both types of imagery to improve the accuracy of FER.
- Our work demonstrates the effective integration of CNN with transfer learning (TL) to enhance model performance. This unique combination allows for a more accurate classification of facial expressions compared to models using only one technique.
- The proposed model was tested on the publicly accessible VIRI and NVIE databases, achieving impressive accuracy rates of 84.44% and 85.20%, respectively. These results surpass the performance of previous methods, showcasing the effectiveness of our approach in real-world applications.
- We conducted an ablation study to evaluate the impact of the attention mechanism on model performance, comparing the results with and without attention across visible, infrared, and combined modalities. This study demonstrates the effectiveness of attention in improving accuracy.
- Additionally, we performed experiments with other baseline models to validate the superiority of our approach. The results showed that our proposed method with attention-enhanced ResNet-18 outperformed these baseline models.

The structure of the article is as follows. "Previous Work" covers the earlier works on the visible and infrared datasets and "Materials" covers the study material. Our proposed work

is explained in "Proposed Models", and the evaluation method is discussed in "Evaluation Method". Experiments and results are covered in "Experiments and Results" while conclusion and future directions are discussed in "Conclusion and Future Directions".

# PREVIOUS WORK

First, there is a section dedicated to the visible dataset; second, there is a section dedicated to the infrared dataset; and third, there is a section dedicated to the combined (visible and infrared) datasets.

## Utilizing the visible dataset for FER

Researchers in *Subarna & Viswanathan (2018)* introduced a deep convolutional spatial neural network (DCSNN). This model was trained on the FER-2013 and JAFFE datasets and was tested in real-time for facial expression detection using a live camera. In order to classify emotions in images, researchers in *Lin & Lee (2018)* proposed a novel optimal technique using CNN and SVM in conjunction with two pretreatment filters, namely brightness and contrast and edge extraction filters.

A CNN leveraging the depth volume of its layers was proposed to tackle the FER problem (*Wang et al., 2018*). This approach employed a CNN with five consecutive convolutional layers, three max pooling layers, one fully connected layer, and an output layer to classify expressions using a softmax classifier. Additionally, a compact CNN with dense connections was introduced to mitigate the overfitting issue that occurs with finite-sized training sets (*Dong, Zheng & Lian, 2018*).

Researchers have developed a real-time framework for identifying driver sentiment using Grassmann manifolds and deep learning (*Verma & Choudhary, 2018*). Deep neural networks (DNNs) and Haar-cascades were employed to locate the region of interest (RoI). Additionally, a study was conducted to compare the accuracies of pre-trained models, including Inception v3, VGG19, VGGFace, and VGG16, with a CNN trained from scratch on publicly available databases such as CK+, JAFFE, and FACES (*Sajjanhar, Wu & Wen, 2018*). The CNN designed from scratch consisted of two convolutional layers, two max-pooling layers, and two dense layers. A real-time FER system utilizing two deep networks, CNN and DNN, on the CK+ database was discussed in *Jung et al. (2015)*. Another DNN was designed to recognize micro-expressions using a transfer learning technique (*Peng et al., 2018*). In this approach, the Active Appearance Model (AAM) was employed to segment the face ROI and normalize the pixels to a size of 224 × 224. A hybrid deep learning model was introduced to effectively capture the characteristics in a video sequence for FER (*Zhang et al., 2019*). While in *Ramalingam & Garzia (2018)*, researchers proposed a transfer learning approach using the deep neural networks VGG16 and VGG19 for expression prediction.

Researchers have developed a reliable and efficient FER system using hierarchical deep neural networks (*Kim et al., 2019*). While in *Zhang et al. (2019)* employed a recurrent neural network (RNN) known as the spatial-temporal RNN (STRNN) to recognize emotions based on facial images and EEG signals, leveraging both spatial and temporal information. To enhance accuracy and develop a system highly sensitive to image details,

the authors in *Wang et al. (2018)* fine-tuned the CASME and Large MPI databases to generate 12 emotion datasets, which were then input into a convolutional block CNN (CBCNN). In another study aimed at analyzing customer satisfaction, researchers in *Ramdhani, Djamal & Ilyas (2018)* proposed a CNN to perform FER on two types of databases: FER2013 and a custom-created database for recognizing four different expressions. Deep Residual Network (ResNet50) was used to measure consumer happiness in video surveillance at the airport (*Sugianto, Tjondronegoro & Tydd, 2018*). Using the FER deep learning method, researchers in *Nugroho, Harmanto & Al-Absi (2018)* demonstrated a smart home system for identifying pain, particularly in older individuals.

In *Wang et al. (2018)*, the authors developed a shallow residual network designed to analyze soldiers' mental states through facial expression recognition, addressing the low accuracy rates commonly seen in conventional FER models. In another study (*Ul Haque & Valles, 2018*), a FER system using a deep convolutional neural network (DCNN) was developed and trained on the FER2013 dataset, specifically for children with autism. Researchers in *Kuo, Lai & Sarkis (2018)* introduced a frame-to-sequence methodology that incorporates temporal information from facial images, making it well-suited for portable devices. In *Dewan et al. (2018)*, the authors proposed an alternative deep learning model featuring two deep belief network (DBN) models, each comprising three hidden layers. In order to circumvent the overfitting issue that arises in numerous real-time applications, a transfer learning technique was employed (*Nguyen et al., 2018*). A unique approach to FER was put forth by the authors in *Wu et al. (2018)* using unique landmark detection networks for different position faces. For landmark detection and FER, Deep CNN with five convolution and two pool layers was employed. A fascinating method was created by the authors in *Li et al. (2018)* to take use of the fact that individuals express themselves in different ways.

Another work introduced a customized VGG-19 (CVGG-19) model that integrates design elements from VGG, Inception-V1, ResNet, and Xception (*Kim, Poulose & Han, 2024*). Using four datasets—JAFFE, CK Plus, KDEF, and AffectNet—researchers achieved high accuracy in emotion classification with the VGG-16 model, employing transfer learning (*Punuri et al., 2024*). Another work presents a fuzzy multimodal fusion network (FMFN) that uses fuzzy logic in multi-feature spaces for recognizing emotions based on facial expressions and conducting gestures (*Han, Chen & Ban, 2024*). To enhance FER system performance in autonomous vehicles, researchers discussed a facial image threshing (FIT) machine that leverages advanced features from a pre-trained facial recognition model and Xception algorithm (*Kim, Poulose & Han, 2021*). The FIT machine refines the dataset by removing irrelevant images, correcting misplaced data, and merging datasets on a large scale, supplemented with data augmentation.

## Utilizing the infrared dataset for FER

A novel model for near-infrared (NIR) facial expression identification was proposed in *Zhang et al. (2020)*. The method uses correlation emotion label distribution learning to learn many emotions associated with each expression based on the similarity of the expressions. Novel approaches to facial emotion identification based on infrared thermal

images were presented in *Assiri & Hossain (2023)*. Firstly, the full facial image is divided into four parts. Only four active regions (ARs) were then approved for the preparation of training and testing datasets. From thermal infrared facial images, researchers were able to learn thermal features for emotion recognition through the use of the deep Boltzmann machine (*Wang et al., 2014*). Using the thermal infrared images, the face is first found and normalized. Next, a two-layer deep Boltzmann machine model was trained.

In order to create a robot that can comprehend human emotions, researchers in *Yoshitomi et al. (2015)* provided a technique for identifying facial expressions. The feature parameter of facial expression was extracted in the region of the mouth and jaw using a two-dimensional discrete cosine transform, and thermal image processing was used to analyze the video.

Using the training data of "taro" for the facial expressions of "angry," "sad," and "surprised," as well as the training data of respective pronunciations for the intentional facial expressions of "happy" and "neutral," researchers in *Nakanishi et al. (2015)* investigated the influence of training data on facial expression recognition accuracy. The proposed method successfully differentiated among the three facial emotions of "happy," "neutral," and "other," achieving an average accuracy of 72.4% across the subjects "taro," "koji," and "tsubasa." A new FER approach was presented in *Poursaberi, Yanushkevich & Gavrilova (2013)*, which extracts features from infrared images using the Gauss-Laguerre (GL) filter on circular harmonic wavelets. Complex texture features can be accurately extracted from an infrared image by creating a set of redundant wavelets using GL filters with suitably adjusted parameters. The interesting field of infrared thermal imaging (IRTI) was illustrated and some of its geometry characteristics were analyzed in order to offer a facial emotion identification approach employing mathematical morphology in *Jiang et al. (2006)*.

Another study, detailed in *Elbarawy, Ghali & El-Sayed (2019)*, explores the use of CNN for FER to analyze human emotional states in thermal images. Unlike traditional methods that rely on feature extraction and extensive pre-processing, CNN, a subset of deep learning techniques, leverages its multi-layer architecture to automatically classify and learn significant features from raw image data. A useful technique for analyzing multi-scale, multi-directional texture changes is the wavelet transform. Consequently, a wavelet transform-based approach for facial expression identification from thermal images was suggested in *Wang, Lv & Wang (2008)*. A method for pose- and illumination-invariant facial emotion recognition utilizing near-infrared camera images and accurate 3D form registration was proposed (*Jeni, Hashimoto & Kubota, 2012*). Constrained local models (CLM) can be used to determine precise 3D shape information of the human face by iteratively fitting a dense model to an unknown sample. For the purpose of FER from infrared images, a deep learning model known as IRFacExNet (InfraRed Facial Expression Network) was proposed in *Bhattacharyya et al. (2021)*. In order to extract dominating features from the input images that are specific to the expressions, it makes use of two building blocks: the residual unit and the transformation unit. The RWTH Aachen University's IR database, a publicly accessible dataset, was used to assess the performance of the suggested model.

### Utilizing the visible and infrared datasets for FER

Recently, the fusion of visible and infrared images has been utilized to enhance recognition performance. This technique extracts critical information from multiple modalities and combines it for recognition purposes (*Zou et al., 2021*).

In a study conducted by researchers in *Wang et al. (2014)*, visible and infrared images were integrated to recognize facial expressions in three stages. Experiments conducted on the natural visible and infrared facial expression (NVIE) spontaneous database (*Wang et al., 2010*) validated the effectiveness of the proposed methodologies, demonstrating the additional benefits of thermal infrared images in identifying visible facial expressions.

Researchers have developed an ensemble-based solution for facial expression recognition by combining voice, infrared, and visible images (*Siddiqui & Javaid, 2020*). This framework was implemented in two levels: the first layer detects emotions using single modalities, while the second layer integrates these modalities to classify emotions. CNNs were utilized for both feature extraction and classification. In *Nguyen et al. (2023)*, a multimodal facial expression database was introduced, comprising thermal infrared and spontaneous visible video recordings. The dataset also included emotional intensity statistics, categorizing each emotion into three levels: low, medium, and high. Seven spontaneous emotions were captured from thirty subjects. Another work was discussed for spontaneous facial expression recognition using feature-level fusion of visible and thermal infrared facial images (*Wang & Wang, 2011*). Tested on the NVIE database, this fusion approach improves accuracy in recognizing negative expressions and reduces discrepancies, enhancing overall expression recognition performance. Researchers extended the work for spontaneous facial expression recognition by fusing features from visible and thermal infrared images (*Wang & He, 2013*). First, active appearance model parameters and head motion features were extracted from visible images, while statistical parameters were extracted from thermal images. A multiple genetic algorithms-based fusion approach was then applied to find the optimal combination of similarity measurements and feature subsets.

## MATERIALS

This section outlines the attention mechanism that is used in this study. The attention mechanism employed in our modified ResNet-18 model plays a crucial role in dynamically weighting the importance of different facial regions, thereby improving the model's ability to capture fine-grained details and context-specific features essential for distinguishing between similar expressions.

### Self-attention

It has been difficult for previous convolution techniques to determine associations with pixels that are located far away. Only specific local fields possess all of the features. Self-Attention (*Vaswani et al., 2017*) was suggested as a solution to these issues. Self-attention computation is comparable to convolution techniques, yet establishing connections with distant areas is simple. A graphic illustration of self-attention is shown in Fig. 2. An input is split into its Query, Key, and Value, all of which have the same shape. In that order, these
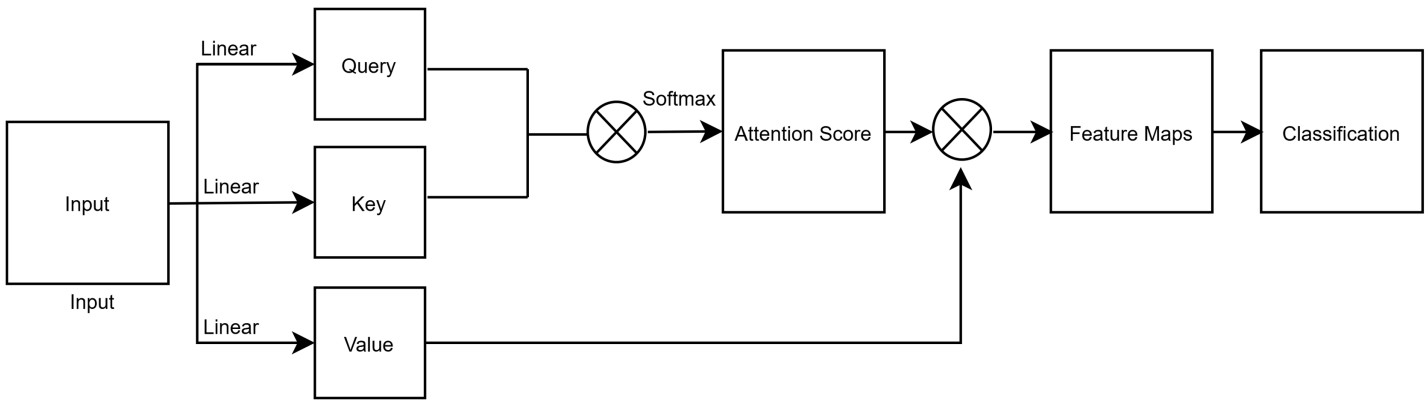

**Figure 2 A figure depicting a process of self-attention.**

represent the mean batch size, query length, and model dimension. We pass the mapped score to the soft-max layer after first obtaining it from the convolution operation of the query and key. The convolution of the Score and Value is the result of the attention operation. Finally, the final output of self-attention is used to perform classification in a fully connected layer.

## PROPOSED MODELS

We first proposed single models for the classification of five expressions available in the VIRI database namely, angry, happy, neutral, sad, and surprised, and three expressions namely happy, fear, and disgust available in the NVIE database. For this purpose, we utilized a modified architecture of ResNet-18 with an attention mechanism. ResNet-18, a variant of deep neural networks, is renowned for its use of skip connections or shortcuts, which help in bypassing certain layers. This architectural modification is particularly advantageous as it addresses the vanishing gradient problem, enabling the network to achieve faster convergence and improved accuracy. The attention mechanism allows the model to focus on important features within the image, enhancing its ability to capture and interpret subtle details in facial expressions. This combination of ResNet-18 and attention aims to improve the model's performance by providing a more nuanced and context-aware analysis of facial expressions. By leveraging this robust model, the research aims to enhance the precision of facial expression classification, effectively distinguishing subtle emotional cues across different imaging modalities. The modified ResNet-18 architecture, augmented with an attention mechanism, provides a powerful tool for analyzing complex datasets and advancing the field of affective computing. Through these single models, we can explore the nuanced differences in emotional expression captured by the VIRI and NVIE databases, contributing to the development of more sophisticated and responsive facial recognition systems.

Secondly, we proposed an early fusion multi-modal approach that merges pair-wise features from the visible and infrared modalities. This method overcomes the limitations of using a single modality, which may not capture enough information for precise image

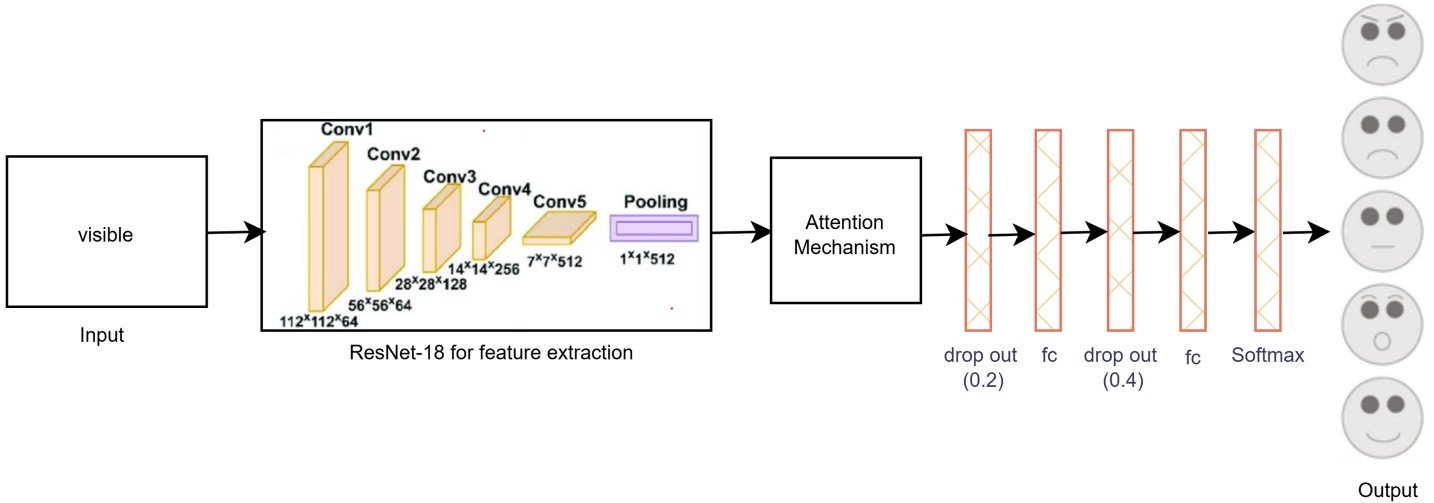

**Figure 3 Proposed single-modal approach for the visible dataset.**

classification. By combining features from both visible and infrared images, the model leverages complementary data, improving its ability to detect subtle details that might be overlooked with a single modality. We utilized the modified ResNet-18 architecture with a self-attention mechanism for this purpose. The CNN's concatenation function is used to integrate these features, forming a more comprehensive representation of the data. The self-attention mechanism further improves the model's focus on the most relevant parts of the images, ensuring that crucial features from both modalities are effectively combined.

This fusion method improves the robustness and accuracy of the classification process, particularly in challenging scenarios where individual modalities fall short. By combining the strengths of both visible and infrared modalities, and utilizing the advanced capabilities of ResNet-18 with self-attention, the model becomes more effective in accurately categorizing facial expressions. This contributes to the development of more reliable and nuanced facial recognition systems.

The proposed single-modal approach for visible modalities employs the ResNet-18 CNN architecture with a self-attention mechanism, as depicted in Fig. 3. In the final layer of ResNet-18, we incorporated two dropout layers, two fully connected (FC) layers, and a SoftMax layer tailored to the visible dataset. Dropout layers are utilized to mitigate overfitting by randomly deactivating a portion of the input units during training, thus enhancing generalization. The FC layers convert the extracted features into a format suitable for classification, while the SoftMax layer ensures that the output probabilities for the various facial expression categories total to one. This architectural adjustment improves the model's capacity to accurately classify facial expressions by effectively learning and differentiating the subtle patterns present in the visible image dataset. These enhancements contribute to making the model robust and reliable for practical use in real-time facial expression recognition systems.

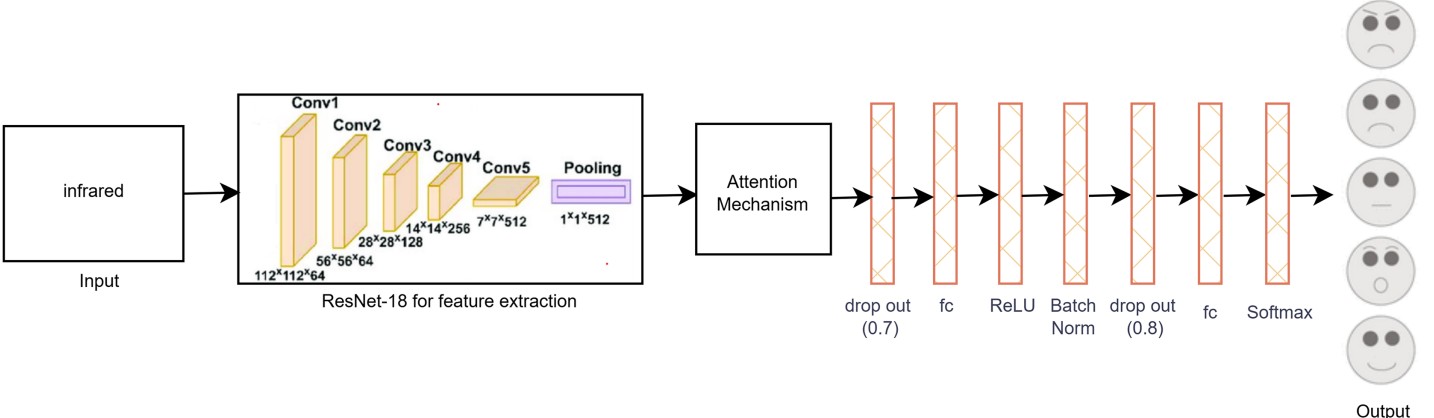

**Figure 4 Proposed single-modal approach for the infrared dataset.**

Because there were few visible images in the dataset, there was some overfitting even if the features could be distinguished with the unaided eye. In order to solve this problem, data augmentation was carried out in order to increase the size of the dataset, producing more diverse examples of the training images and enhancing the generalization capacity of the model. Further efforts were made to reduce overfitting by means of parameter tuning, which entailed methodically modifying hyper parameters under strict observation of accuracy and loss on both the training and validation sets. This iterative process ensured that the model achieved optimal performance without compromising its ability to generalize to new data.

As a result of these efforts, the final layer of the ResNet-18 model was modified, as shown in Fig. 3. These modifications included the addition of two dropout layers to reduce overfitting, two FC layers to enhance feature representation, and a SoftMax layer to ensure proper classification probabilities. These adjustments helped in fine-tuning the model, leading to improved accuracy and robustness in classifying facial expressions from visible images.

Figure 4 illustrates the proposed single-modal approach for the infrared dataset utilizing the modified ResNet-18 CNN architecture with a self-attention mechanism. The final layer of ResNet-18 was adapted to the infrared dataset by adding two dropout layers, an activation function, a batch normalization layer, three fully connected (FC) layers, and a SoftMax layer. These modifications were essential to accommodate the unique properties of infrared images. Unlike visible images, many infrared images exhibit less distinct features, which introduces additional challenges in accurately classifying facial expressions. To overcome the limitations posed by the small dataset size and to enhance model performance, various data augmentation techniques were applied to increase the diversity and quantity of infrared images, thereby improving the model's training effectiveness.

To reduce overfitting, we carefully monitored the loss and accuracy metrics for both the training and validation sets during the hyper parameter tuning phase. This process

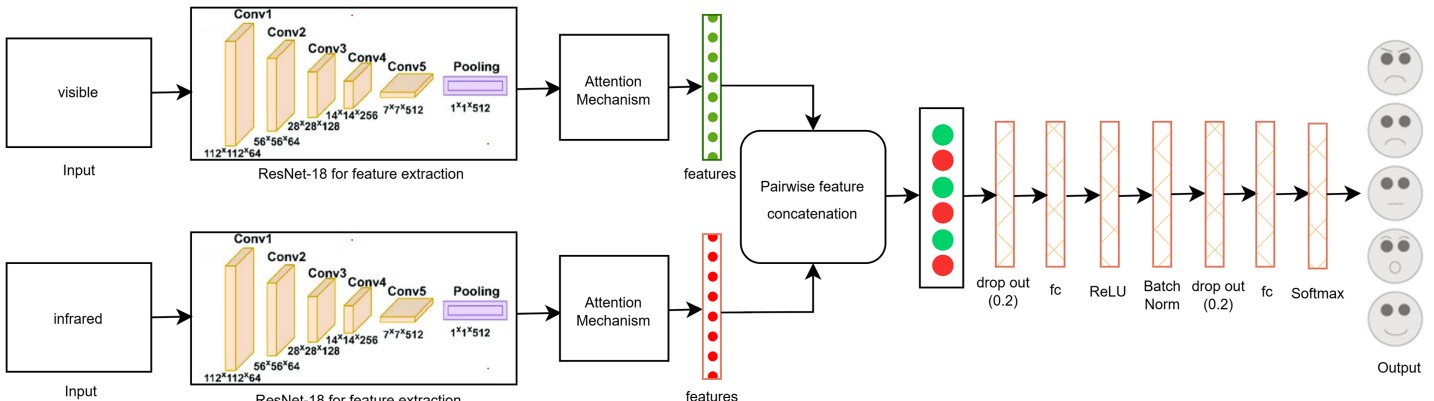

**Figure 5 Proposed multi-modal approach using early fusion method by concatenating the features pair-wise from visible and infrared datasets.**

involved iterative adjustments of the model's hyper parameters to achieve optimal performance. Consequently, the final layer of the ResNet-18 model was adapted, as depicted in Fig. 4, to better accommodate the specific characteristics of the infrared dataset. These modifications enhanced the model's capacity to generalize and accurately classify facial expressions in infrared images, thereby improving the overall robustness and reliability of the proposed method.

Figure 5 shows the proposed multi-modal early fusion method for pair-wise concatenation of features from the visible and infrared datasets using the ResNet-18 architecture with the self-attention mechanism. This approach leverages the strengths of both modalities to enhance the accuracy and robustness of facial expression classification. In order to gather valuable data that may be utilized for image categorization, feature extraction was done. For example, if we have an image of a human face, feature extraction allows us to identify its mouth, nose, and eyes. The strengths of each particular model can be combined by concatenating information from other modalities, which improves accuracy overall. Because it integrates data from several modalities, its fusion design greatly improves system efficiency, resilience, and flexibility. The model can capture more detailed and subtle information because of the combination of visible and infrared features, which is essential for precise classification.

In this method, we collected features from both modalities using ResNet-18 before the last FC layer, ensuring that rich information from both visible and infrared modalities is utilized. These features were then concatenated pair-wise to form a unified feature vector. To further refine the model and prevent overfitting, we added two dropout layers, one batch normalization layer, one activation layer, three FC layers, and a SoftMax layer. The dropout layers help in regularization by randomly dropping units during training, the batch normalization layer stabilizes and accelerates the training process, and the fully connected layers transform the concatenated features into a format suitable for classification. The SoftMax layer at the end ensures that the output probabilities across the different facial expression classes sum to one.

This comprehensive fusion approach, integrating self-attention mechanisms and modified ResNet-18 architecture, allows for more effective learning from the combined datasets, ultimately leading to a more reliable and accurate FER system.

## EVALUATION METHOD

To comprehensively evaluate the performance of our proposed facial expression recognition model, we employed several standard metrics and performed extensive experiments on two publicly accessible datasets: VIRI (*Siddiqui & Javaid, 2020*; https://www.yazdan.us/research/repository) and NVIE (*Wang et al., 2010*; https://ustc-ac.github.io/nvie/).

### Databases

- **VIRI database** (https://www.yazdan.us/research/repository): Contains 566 image pairings of five distinct facial expressions (angry, happy, neutral, sad and surprised) captured in visible and infrared modalities.
- **NVIE database** (https://ustc-ac.github.io/nvie/): Contains visible and infrared images from over 100 subjects, capturing six fundamental facial expressions (happy, sad, surprised, fear, rage, disgust).

### Experimental setup

- **Data splitting**: Each dataset was divided into training (70%), validation (15%), and test (15%) sets. Data augmentation techniques such as rotations, zooming, distortion, shear, and flipping were applied to the training sets to increase the diversity of the training examples and mitigate overfitting.
- **Training**: The ResNet-18 model, modified with an attention mechanism, was trained separately on visible and infrared images, as well as on a combined dataset using an early fusion approach. We utilized the Adam optimizer with a CosineAnnealingLR scheduler and set the batch size to 64. The loss function used for training was cross-entropy.

## EXPERIMENTS AND RESULTS

### Experimental databases

#### VIRI database

The VIRI database serves as a valuable resource for studies involving visible, infrared, and MSX image modalities, providing 566 image pairings taken in uncontrolled, naturalistic environments (*Siddiqui & Javaid, 2020*). This diversity in background settings is crucial for testing the robustness and versatility of facial recognition systems in real-world conditions. The database features images from 110 individuals, capturing five distinct facial expressions like, angry, happy, neutral, sad, and surprised. Each image maintains a consistent resolution of 500 × 500 pixels, ensuring uniformity in data analysis.

The differences between visible and infrared images highlight the challenges in facial feature detection, particularly in discerning finer details such as the inner brow in infrared images. This complexity underscores the need for advanced algorithms capable of

accurately interpreting facial expressions across varying image modalities, contributing significantly to the development of more sophisticated and reliable facial recognition technologies.

### NVIE database

The NVIE spontaneous database is a comprehensive resource for facial expression analysis, containing a wide variety of data points from over 100 subjects (*Wang et al., 2010*). It includes both visible and infrared images, which are crucial for enhancing the accuracy and robustness of facial recognition systems under different lighting conditions. The database is meticulously curated to encompass six fundamental facial expressions namely, happy, sad, surprised, fear, rage, and disgust, providing a rich dataset for emotion recognition research. Several examples from the NVIE database illustrate the diversity and spontaneity of the captured expressions, which are essential for training and validating machine-learning models. This dataset is instrumental in advancing the field of affective computing and developing more empathetic and responsive human-computer interaction systems.

## Data preprocessing

The VIRI database, which is publicly available, contains 111 samples for angry expressions, 114 samples for happy, 114 samples for neutral, and 113 samples for surprised expressions. We divided the images for each modality in the VIRI database into three sets: training set, validation set, and test set, as illustrated in Fig. 6. The split ratios were 70% for training, 15% for validation, and 15% for testing. To train the ResNet-18 for expression recognition using the VIRI database, a substantial number of images was necessary to achieve the desired level of accuracy. Therefore, to augment the datasets, various image augmentation techniques such as rotations, zooming, distortion, shear, and flipping were applied to each training set for each modality. For instance, for the angry expression, the total number of training images before augmentation was 77 (as shown in Fig. 6A). By applying these augmentations, we increased the size of the training dataset for the angry expression accordingly as shown in Fig. 6B.

- To start, we enhanced each image by applying the rotation augmentations five times, resulting in a total of ($77 \times 5 = 385$) images.
- Second, we enhanced each image by applying the zooming augmentation five times, resulting in a total of ($77 \times 5 = 385$) images.
- Third, we enhanced each image by applying the distortion augmentation five times, resulting in a total of ($77 \times 5 = 385$) images.
- Fourth, we enhanced each image by applying the shear augmentation five times, resulting in a total of ($77 \times 5 = 385$) images.
- In the last, we enhanced each image by applying the flipping augmentation five times, resulting in a total of ($77 \times 5 = 385$) images.

In total, we obtained 1,925 images (385 images per expression for five expressions) after applying augmentations. We then added the original training images for each expression

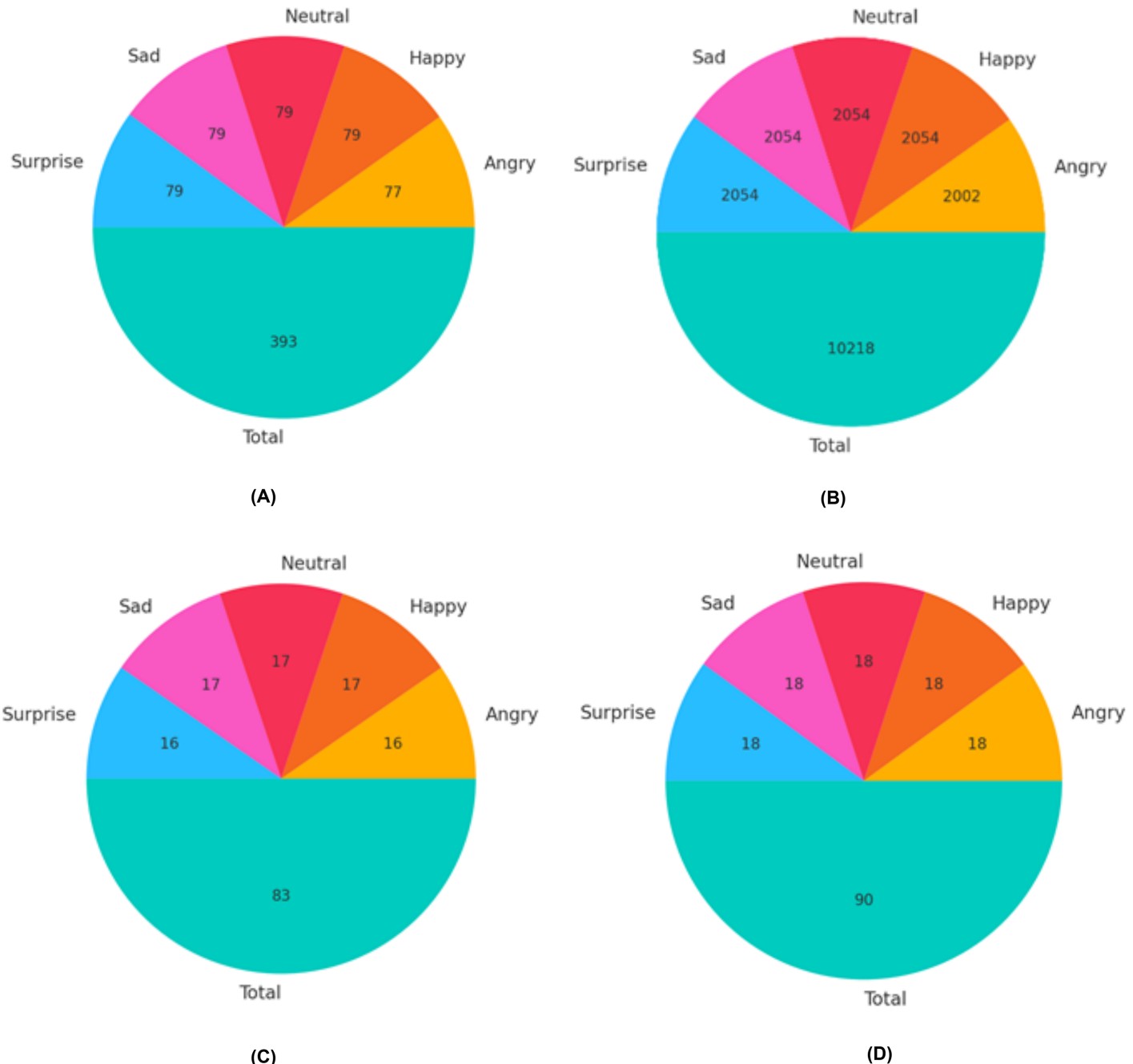

**Figure 6 Distribution of images for training, validation, and test sets in the VIRI database.** (A) Training images before augmentations (B) Training images after augmentations (C) Validation set distribution (D) Test set distribution.

(*e.g.*, 77 images for the angry expression), resulting in 2,002 images for training in the case of the angry expression, as depicted in Fig. 7B. This procedure was repeated to increase the number of images for each expression in the VIRI database. The validation and test sets are depicted in Figs. 6C and 6D. A similar approach was utilized to expand the training set for the NVIE database. The test set was exclusively used to evaluate the model's performance,

while the validation set was employed to fine-tune the network's hyperparameters during the training of the modified ResNet-18 model with an attention mechanism.

## Experimental results

This section outlines our observations and experimental results, detailing the datasets, model configurations, and training parameters to ensure transparency and reproducibility. For the VIRI database, we utilized the original 566 samples as provided by *Siddiqui & Javaid (2020)*, ensuring that our results could be directly compared with previous studies. By employing the same samples, we maintained consistency with established benchmarks, allowing us to effectively assess the improvements introduced by our proposed model. This approach ensured that any observed performance enhancements were due to the model's architecture and techniques rather than variations in data.

In the case of the NVIE database, we followed specific selection criteria to create a balanced and comprehensive dataset. Based on guidelines from *Wang et al. (2014)*, we selected samples with a mean intensity greater than 1 for each expression label, focusing on the three most frequently elicited expressions in the NVIE database—disgust, fear, and happiness. Additionally, each sample included both visible and infrared images, ensuring multi-modal data coverage. This process yielded 518 samples from 123 individuals across varying lighting conditions, providing a robust dataset for evaluating our model's performance in a variety of realistic scenarios. This careful sample selection enabled us to test the model's ability to generalize across diverse expressions and environmental conditions.

For model optimization, we employed the Adam optimizer with a CosineAnnealingLR scheduler across both single and multi-modal experiments. These choices were made to balance efficient convergence and generalization. We set a batch size of 64 for all experiments, which was optimal for our hardware setup, leveraging the memory and processing power of an NVIDIA GeForce RTX A6000 with 48 GB of RAM. Cross-entropy was selected as the loss function for each modality; a common choice for classification tasks as it penalizes incorrect predictions more heavily, encouraging faster convergence. This metric, also known as log loss, quantifies the model's confidence and accuracy by assigning higher penalties as predicted probabilities deviate from actual labels, which is crucial for fine-grained expression classification.

Single-modal models were trained over 50 epochs, while the multi-modal model was trained for 100 epochs, allowing for more in-depth feature learning across combined modalities. All models were implemented using PyTorch, taking advantage of its flexibility and efficiency for deep learning tasks. Our methodological rigor, including consistency with benchmark datasets, careful sample selection, and optimized training configurations, supports the validity and robustness of our results.

### Utilizing the visible dataset

The confusion matrix shown in Fig. 7, now reflecting an accuracy of 83.33%, highlights the results of the fine-tuned ResNet-18 model with an attention mechanism. For the angry expression, 14 samples were accurately classified, while three were misclassified as neutral

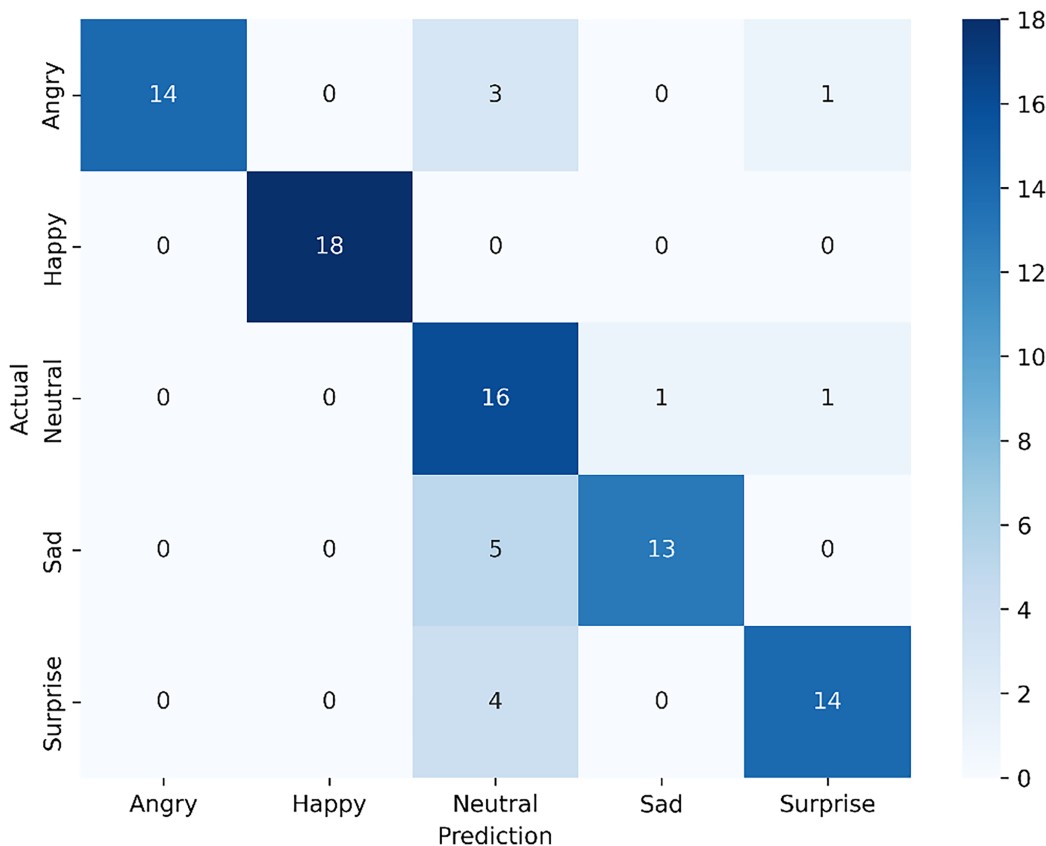

**Figure 7** **Confusion matrix for the proposed single-modal approach using the visible dataset from the VIRI database.**

and one as surprise. The happy expression achieved perfect classification, with all 18 samples correctly identified, demonstrating the model's strong performance in recognizing this emotion. For neutral, 16 samples were correctly classified, with one misclassified as angry and another as surprise. In the sad category, 13 samples were correctly identified, though five were misclassified as neutral, indicating some overlap between these emotions. Lastly, for surprise, the model correctly classified 14 samples but misclassified four as neutral. Despite the fine-tuning applied to ResNet-18, which improved overall accuracy, there remain slight challenges in distinguishing similar expressions, such as sad and neutral or angry and surprise. This suggests that while the model is highly effective, especially for expressions like happy, further exploration of attention mechanisms or additional data augmentation may help enhance its nuanced differentiation capabilities.

### *Utilizing the infrared dataset*

The confusion matrix in Fig. 8 illustrates the performance of our proposed single-modal approach applied to the infrared dataset from the VIRI database, achieving an overall accuracy of 51.11%. For the angry expression, only five samples were correctly identified, while misclassifications occurred with two as happy, three as neutral, seven as sad, and one as surprise. In the happy expression, the model successfully classified 11 samples correctly, with misclassifications of four as angry, two as sad, and one as surprise. For the neutral

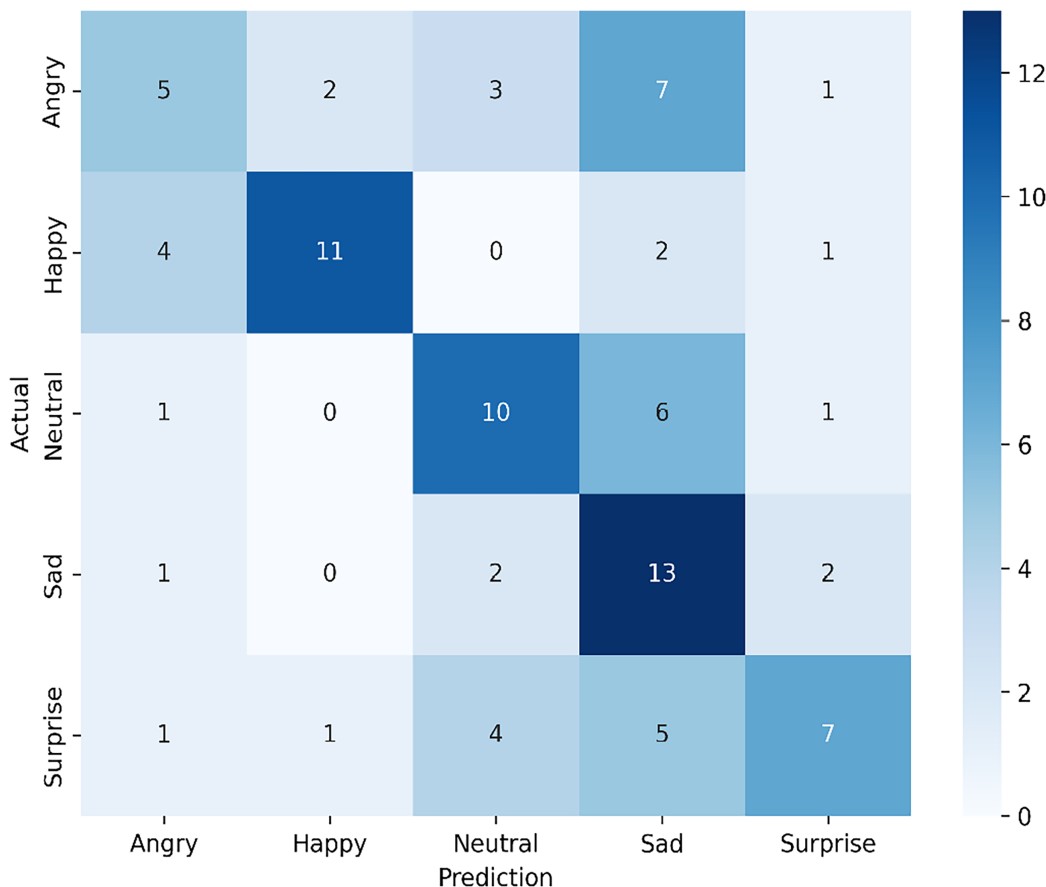

**Figure 8  Confusion matrix for the proposed single-modal approach using the infrared dataset from the VIRI database.**

expression, 10 samples were correctly identified, though one was misclassified as angry, six as sad, and one as surprise. In the sad expression category, 13 samples were accurately classified, with one misclassified as angry, two as neutral, and two as surprise. For surprise, seven samples were correctly identified, while misclassifications occurred with one as angry, one as happy, four as neutral, and five as sad.

This analysis shows that the model demonstrates a relatively high recognition rate for happy, neutral, and sad expressions but struggles with distinguishing angry and surprise expressions. The frequent misclassification of angry as sad and surprise suggests potential overlap in feature representation, which may be due to the nature of the infrared dataset. The variability in classification performance highlights areas for potential improvement, such as incorporating additional tuning specific to infrared data or exploring more robust feature extraction techniques to enhance the distinction between closely related emotional expressions.

### Utilizing the visible and infrared datasets

The confusion matrix in Fig. 9 shows the classification results of our proposed multi-modal approach, which combines visible and infrared modalities from the VIRI database, achieving an overall accuracy of 84.44%. For the angry expression, 12 samples were

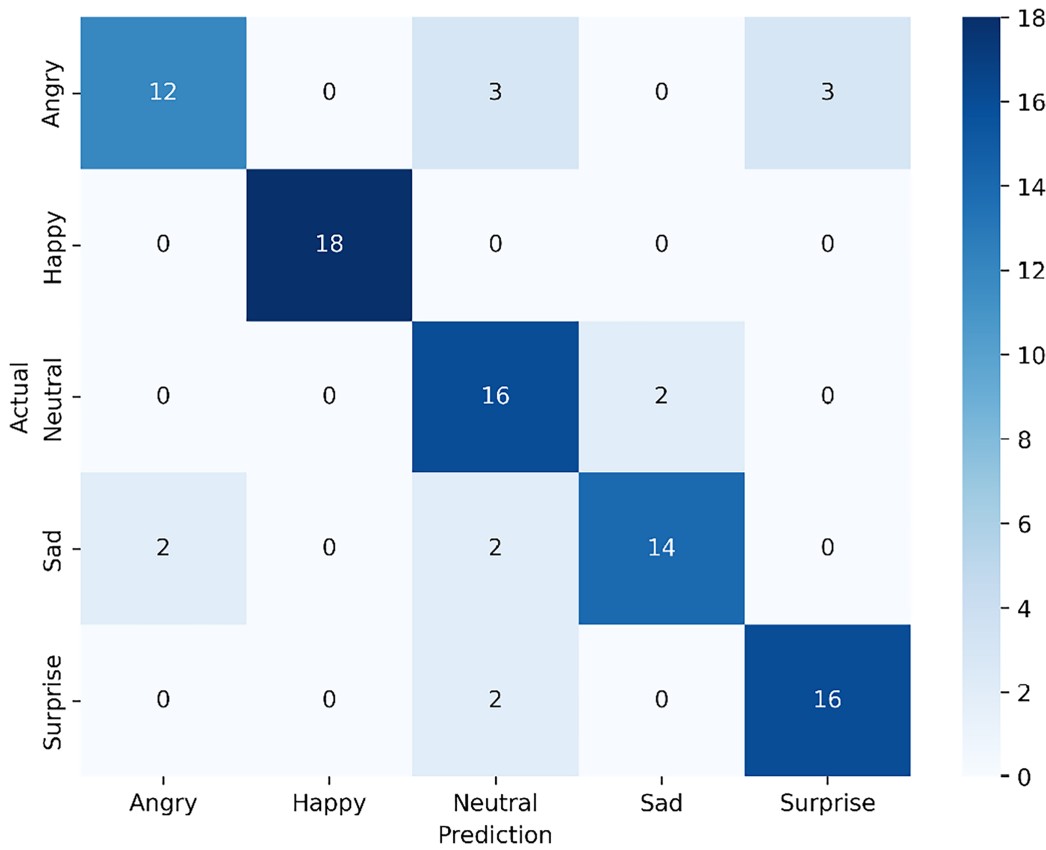

**Figure 9 Confusion matrix for the proposed multi-modal approach using visible and infrared datasets from the VIRI database.**

correctly classified, with three misclassified as neutral and another three as surprise. The happy expression achieved perfect classification, with all 18 samples correctly identified, indicating that the model effectively recognizes this expression in the multi-modal setting. For the neutral expression, 16 samples were accurately classified, though two were misclassified as sad. In the case of sad expression, 14 samples were correctly identified, with minor misclassification—two samples were classified as neutral. Lastly, for surprise, 16 samples were correctly classified, with two misclassified as neutral.

The results indicate that the multi-modal approach provides substantial improvement in overall accuracy compared to single-modal models, especially in recognizing happy and surprise expressions, which were classified with high accuracy. The multi-modal system shows resilience against certain misclassifications seen in the single-modal infrared approach, suggesting that combining visible and infrared data enhances the model's ability to capture more distinguishing features of each expression. However, there is still some overlap between angry and surprise, as well as neutral and sad, which could be areas for further refinement in the model to reduce these misclassifications.

Figure 10 shows the training and validation accuracy and loss curves for the combined visible and infrared datasets from the VIRI database with using a CosineAnnealingLR scheduler over 100 epochs. In the training accuracy (green line, A), the model quickly

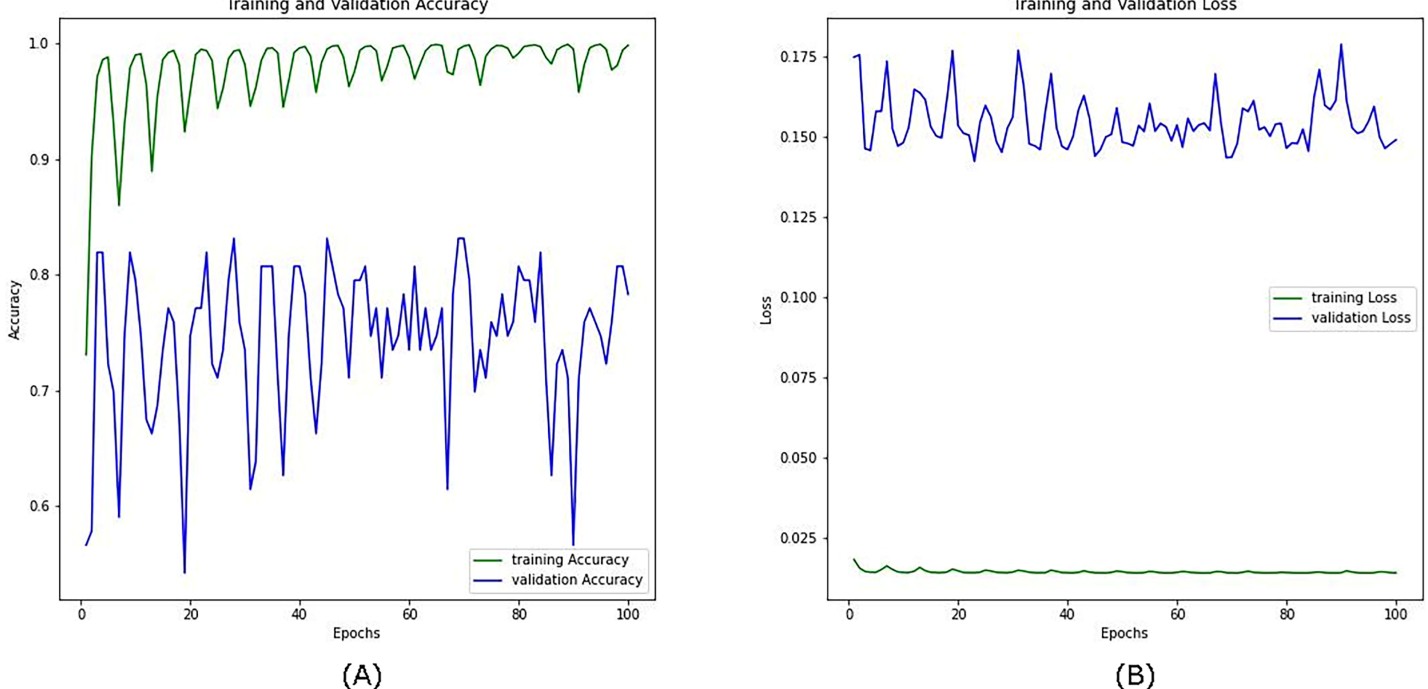

**Figure 10 Accuracy (A) and loss (B) curves for the combined datasets; visible infrared from the VIRI database using CosineAnnealingLR scheduler.**

converges close to 1.0, indicating high accuracy on the training data. However, the validation accuracy (blue line) shows considerable fluctuation, with values ranging between approximately 0.6 and 0.85, suggesting variability in the model's performance on unseen data. This oscillation in validation accuracy implies that while the model learns well from the training data, it struggles to maintain consistent performance across the validation set, likely due to the complexity of integrating features from both visible and infrared modalities.

On the right, the training and validation loss curves further support this observation. The training loss (green line) in Fig. 10B remains low and stable throughout the epochs, indicating effective minimization of error on the training set. In contrast, the validation loss (blue line) fluctuates at a higher level without a clear downward trend, suggesting that the model has not fully generalized to the validation data and may still be overfitting despite the benefits of multi-modal data. Another reason we anticipate is the presence of sharp minima (*You, Gitman & Ginsburg, 2017*).

Figure 11 illustrates the performance of the same model on the combined visible and infrared datasets without using the CosineAnnealingLR scheduler. In this case, the training accuracy (green line, A) still increases rapidly and stabilizes at a high value, reflecting effective learning from the training data. However, the validation accuracy (blue line) fluctuates less significantly compared to the previous figure, staying within the 0.6–0.85 range, though still showing variability. This suggests that, although the removal of the

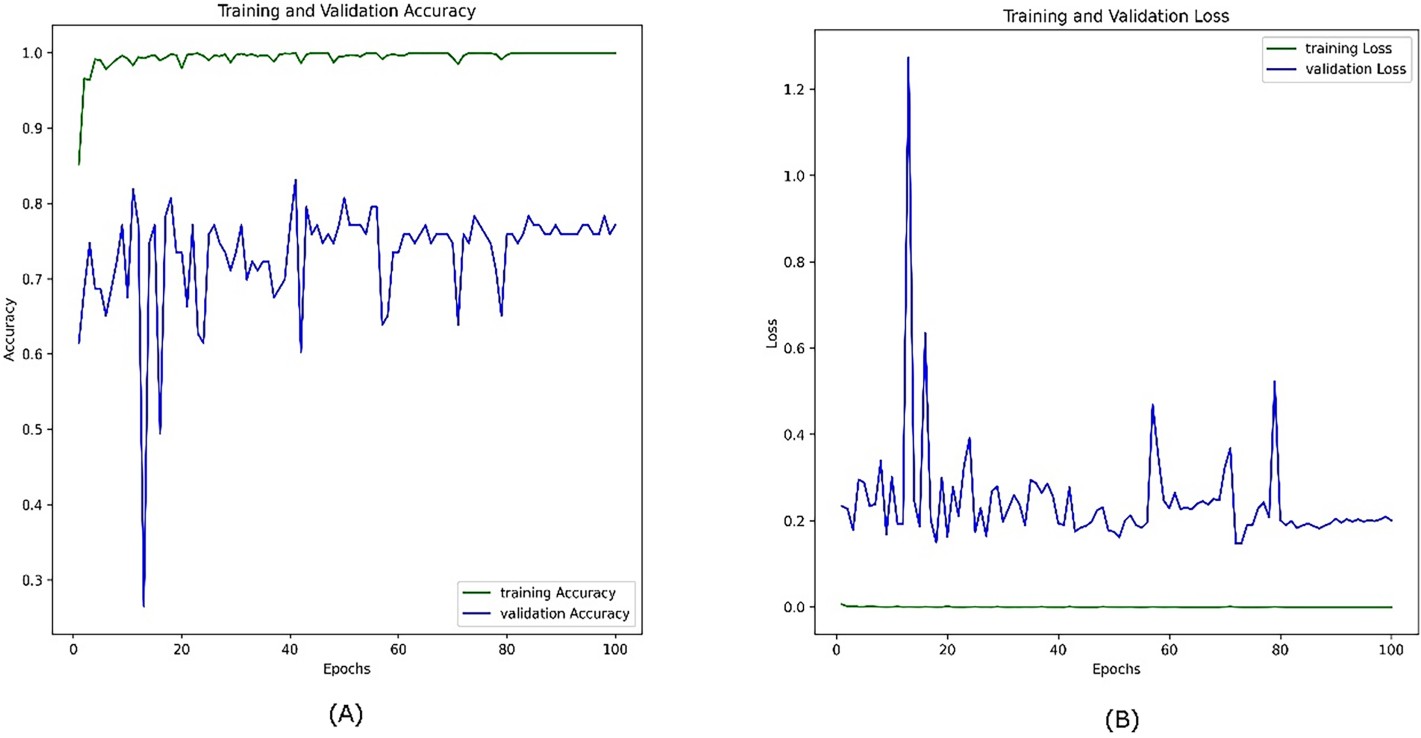

**Figure 11 Accuracy (A) and loss (B) curves for the combined datasets; visible infrared from the VIRI database after removing the CosineAnnealingLR scheduler.**

scheduler reduced some fluctuations, the model still struggles with consistent generalization, likely due to the challenges inherent in the small and diverse dataset. Furthermore, the presence of sharp minima (*You, Gitman & Ginsburg, 2017*) is another anticipated reason.

The training loss (green line, B) remains consistently low and stable, while the validation loss (blue line, B) shows a less erratic pattern, but still oscillates around a relatively higher value. This indicates that while the model's generalization improves without the CosineAnnealingLR scheduler, the dataset size and feature distribution continue to limit the model's full generalization capability. This reinforces the need for a larger and more diverse dataset to achieve better stability in validation performance.

The combined visible and infrared dataset presents additional challenges due to the distinct feature distributions of the two modalities. While data fusion may provide complementary information, the relatively small dataset size and limited diversity continue to contribute to validation instability. Larger, more diverse datasets and advanced fusion techniques will be crucial for improving the stability and generalization of models trained on combined modalities.

### Ablation study

Table 1 presents the results of an ablation study comparing accuracy across different modalities (visible, infrared, and combined visible-infrared) with and without the attention

**Table 1 Ablation study showing accuracy across different modalities.**

| Modality | VIRI | | NVIE | |
|---|---|---|---|---|
| | Without attention (*Naseem, Lee & Kim, 2024*) | With attention | Without attention (*Naseem, Lee & Kim, 2024*) | With attention |
| Visible | 81.11 | 83.33 | 82.71 | 83.94 |
| Infrared | 50 | 51.11 | 55.56 | 57.23 |
| Visible and infrared | 83.33 | 84.44 | 84 | 85.41 |

mechanism for the VIRI and NVIE datasets. This study highlights the impact of the attention mechanism on model performance. For the VIRI dataset, the visible modality achieved an accuracy of 81.11% without attention, which improved to 83.33% when attention was incorporated, indicating that the attention mechanism enhances the model's focus on salient features, leading to a 2.22% increase in accuracy. In the infrared modality, accuracy was relatively low at 50% without attention, and the addition of attention improved it to 51.11%, suggesting limited improvement likely due to inherent challenges in extracting discriminative features from infrared data alone. When combining both visible and infrared modalities, the model achieved 83.33% accuracy without attention and improved to 84.44% with attention, showing that the fusion of modalities with attention yields the highest accuracy for the VIRI dataset.

For the NVIE dataset, similar trends are observed. The visible modality achieved 82.71% accuracy without attention, which increased to 83.94% with attention, reflecting a positive effect of attention on visible data. For the infrared modality, accuracy improved from 55.56% to 57.23% with attention, indicating a slightly higher improvement than in the VIRI dataset, though still relatively modest. Combining visible and infrared modalities resulted in 84% accuracy without attention and 85.41% with attention, again showing that the multi-modal approach with attention provides the best performance.

Overall, this ablation study demonstrates that incorporating the attention mechanism improves accuracy across all modalities, particularly when using the combined visible-infrared data. The results suggest that the attention mechanism helps the model focus more effectively on relevant features, particularly in multi-modal settings, which results in better performance across both datasets.

### Comparison with existing work

In Table 2, our proposed single-modal and multi-modal models show substantial improvements over previous approaches on the VIRI database, particularly when compared with references *Siddiqui & Javaid (2020)* and *Naseem, Lee & Kim (2024)*. For the visible modality, our proposed model achieved an accuracy of 83.33%, which is an improvement of 12.14% over the accuracy of 71.19% reported in *Siddiqui & Javaid (2020)*. This increase highlights the effectiveness of our attention-enhanced model in capturing facial expression features in visible data. In comparison to *Naseem, Lee & Kim (2024)*, which achieved an accuracy of 81.11%, our model shows a modest improvement of 2.22%.

**Table 2  Comparison of our proposed single-modals and multi-modal with the VIRI database.**

| Modality | Accuracy (%) | Precision | Recall | F1-score |
|---|---|---|---|---|
| Visible (*Siddiqui & Javaid, 2020*) | 71.19 | 0.78 | 0.71 | 0.75 |
| Infrared (*Siddiqui & Javaid, 2020*) | 77.34 | 0.78 | 0.77 | 0.78 |
| Visible and infrared (*Siddiqui & Javaid, 2020*) | 82.26 | 0.85 | 0.82 | 0.83 |
| Visible (*Naseem, Lee & Kim, 2024*) | 81.11 | 0.84 | 0.81 | 0.82 |
| Infrared (*Naseem, Lee & Kim, 2024*) | 50 | 0.54 | 0.50 | 0.51 |
| Visible and infrared (*Naseem, Lee & Kim, 2024*) | 83.33 | 0.85 | 0.83 | 0.84 |
| Our visible | 83.33 | 0.86 | 0.83 | 0.84 |
| Our infrared | 51.11 | 0.54 | 0.51 | 0.51 |
| Our visible and infrared | 84.44 | 0.87 | 0.84 | 0.85 |

For the infrared modality, our model reached an accuracy of 51.11%, which is slightly higher than the 50% reported in *Naseem, Lee & Kim (2024)*, showing a 1.11% increase. However, this is lower than the 77.34% achieved in *Siddiqui & Javaid (2020)*, indicating that while our model shows improvement over some approaches, there is still room for refinement when handling infrared data.

In the combined visible and infrared modality, our proposed multi-modal model achieved an accuracy of 84.44%, which represents a 2.18% improvement over the 82.26% reported in *Siddiqui & Javaid (2020)*. This surpasses (*Naseem, Lee & Kim, 2024*) as well, which achieved 83.33%, showing a gain of 1.11%. These enhancements underscore the effectiveness of our multi-modal approach, leveraging both visible and infrared data to produce a more accurate and robust emotion recognition model. This analysis illustrates the significant gains made by our model across different modalities, with the most notable improvements observed in the visible and multi-modal configurations.

In summary, our proposed model demonstrates substantial improvements in the visible and multi-modal settings, outperforming both previous works (*Siddiqui & Javaid, 2020*) and *Naseem, Lee & Kim (2024)*. These enhancements highlight the effectiveness of our model design, particularly in multi-modal applications where combining visible and infrared data allows for a more comprehensive understanding of emotional expressions. The results suggest that our approach provides a more robust and accurate solution for emotion recognition in the VIRI database, though further work could aim to improve performance for infrared data alone.

Table 3 presents the performance of various baseline models—AlexNet, ShuffleNet, MobileNet-v2, DenseNet-201, and VGG-16—when utilizing the attention mechanism on the VIRI database across visible, infrared, and combined visible-infrared modalities. The results are evaluated using metrics such as accuracy, precision, recall, and F1-score.

For the visible modality, MobileNet-v2 achieves the highest accuracy at 80%, with precision, recall, and F1-score values of 0.80, 0.80, and 0.79, respectively, indicating robust performance among the baselines for visible data. DenseNet-201 follows closely with an accuracy of 77.8% and precision, recall, and F1-score values of 0.81, 0.77, and 0.78,

**Table 3 Experimental results for other baselines utilizing the attention mechanism with the VIRI database.**

| Models | Visible | | | | Infrared | | | | Visible and infrared | | | |
|---|---|---|---|---|---|---|---|---|---|---|---|---|
| | Accuracy (%) | Precision | Recall | F1-score | Accuracy (%) | Precision | Recall | F1-score | Accuracy (%) | Precision | Recall | F1-score |
| AlexNet | 38.89 | 0.39 | 0.38 | 0.38 | 26.51 | 0.22 | 0.27 | 0.22 | 40.22 | 0.41 | 0.42 | 0.42 |
| ShuffleNet | 76.67 | 0.77 | 0.77 | 0.76 | 40 | 0.39 | 0.40 | 0.38 | 81.11 | 0.81 | 0.81 | 0.81 |
| MobileNet-v2 | 80 | 0.80 | 80 | 0.79 | 48.89 | 0.55 | 0.49 | 0.49 | 82.22 | 0.82 | 0.81 | 0.81 |
| DenseNet-201 | 77.8 | 0.81 | 0.77 | 0.78 | 43.33 | 0.56 | 0.43 | 0.44 | 80 | 0.80 | 0.84 | 0.80 |
| Vgg-16 | 36.67 | 0.36 | 0.36 | 0.35 | 25 | 0.56 | 0.26 | 0.25 | 38.44 | 0.40 | 0.44 | 0.41 |

respectively, suggesting good generalization for visible images. In contrast, AlexNet and VGG-16 perform poorly, with accuracies of 38.89% and 36.67%, respectively, indicating limited effectiveness in recognizing emotions using visible data.

In the infrared modality, MobileNet-v2 again leads with an accuracy of 48.89%, precision of 0.50, recall of 0.48, and F1-score of 0.49, indicating that it is comparatively better suited for infrared data among the baselines. DenseNet-201 performs next best with an accuracy of 43.33% and other metrics indicating moderate capability. Both AlexNet and VGG-16 perform poorly in the infrared setting, with accuracies of 26.51% and 25%, respectively, showing their limitations in extracting useful features from infrared data.

For the combined visible and infrared modalities, MobileNet-v2 achieves the highest accuracy of 81.11%, with precision, recall, and F1-score values around 81, showing effective fusion of both modalities. DenseNet-201 also performs well with an accuracy of 81.11% and similar metrics. The attention mechanism enables these models to leverage complementary information from both modalities effectively. AlexNet and VGG-16, however, continue to perform poorly, with accuracies of 40.22% and 40%, respectively, suggesting that they do not benefit significantly from multi-modal data even with attention.

In summary, MobileNet-v2 and DenseNet-201 show the best performance across all modalities, particularly in multi-modal settings, highlighting their effectiveness when combined with attention mechanisms. These results suggest that lightweight architectures like MobileNet-v2, with optimized feature extraction, perform well in emotion recognition tasks on the VIRI database, while larger architectures like AlexNet and VGG-16 may not be well suited for this dataset and task.

In Table 4, our proposed visible model, with an accuracy of 83.94%, shows clear improvements over previous works in the NVIE database. Specifically, it surpasses the visible accuracy in *Naseem, Lee & Kim (2024)* by 1.23%, in *Wang et al. (2014)* by 8.61%, and in *Wang et al. (2010)* by 5.13%. This improvement underscores the effectiveness of our model in leveraging attention mechanisms to enhance feature capture from visible data. In the infrared modality, our model achieved an accuracy of 57.23%, exceeding (*Naseem, Lee & Kim, 2024*) by 1.67%, (*Wang et al., 2014*) by 4.33%, and (*Wang et al., 2010*) by 3.29%. These results demonstrate that our approach enhances feature extraction even in more challenging modalities like visible and infrared.

**Table 4 Comparison of our proposed single-modals and multi-modal with the NVIE database.**

| Modality | Accuracy (%) |
|---|---|
| Visible (*Wang et al., 2014*) | 75.33 |
| Infrared (*Wang et al., 2014*) | 52.90 |
| Visible and infrared (*Wang et al., 2014*) | 76.82 |
| Visible (*Naseem, Lee & Kim, 2024*) | 82.71 |
| Infrared (*Naseem, Lee & Kim, 2024*) | 55.56 |
| Visible and infrared (*Naseem, Lee & Kim, 2024*) | 84 |
| Visible (*Wang et al., 2010*) | 78.81 |
| Infrared (*Wang et al., 2010*) | 53.94 |
| Visible and infrared (*Wang & Wang, 2011*) | 61.10 |
| Visible and infrared (*Wang & He, 2013*) | 63.20 |
| Our visible | 83.94 |
| Our infrared | 57.23 |
| Our visible and infrared | 85.41 |

For the combined visible and infrared modality, earlier approaches reported a range of accuracies: 61.10% (*Wang & Wang, 2011*), 63.20% (*Wang & He, 2013*), and 84% (*Naseem, Lee & Kim, 2024*) being the highest. Our proposed multi-modal model achieved an accuracy of 85.41%, surpassing the best previous result by 1.41% compared to *Naseem, Lee & Kim (2024)*, 24.31% compared to *Wang & Wang (2011)*, and 22.21% compared to *Wang & He (2013)*. This improvement emphasizes the strength of our fusion method, which effectively leverages complementary features from both visible and infrared data while using attention to focus on the most relevant aspects of each. The consistency of improvements across all modalities and datasets illustrates the robustness of our model, confirming that our enhancements in attention-based feature selection and fusion yield significant benefits in emotion recognition accuracy across varying conditions and datasets.

In summary, Table 4 demonstrates that our proposed models outperform existing methods across all modalities, particularly in the multi-modal setting, where we achieve the highest accuracy. These findings validate our approach, displaying the advantages of our attention-enhanced fusion model in integrating visible and infrared data for improved emotion recognition accuracy.

### Limitations of our model due to the dataset

While the proposed model demonstrates strong training performance and competitive accuracy across the visible, infrared, and combined visible-infrared datasets, there are several limitations. These limitations stem from dataset constraints, modality-specific challenges, and the inherent complexities of multi-modal data integration.

- **Visible dataset:** The relatively small size of the visible dataset, comprising 550 images, limits the diversity and variability necessary for robust generalization. Although data

augmentation was applied to expand the dataset, the augmented images do not introduce fundamentally new features. This constraint contributes to the observed fluctuations in validation accuracy and loss curves, highlighting the need for larger, more diverse datasets.

- **Infrared dataset:** The infrared dataset presents additional challenges due to the inherent nature of infrared images, which lack detailed features such as texture and contrast that are critical for distinguishing subtle expressions. These challenges, compounded by the dataset's small size, lead to instability in validation performance, as reflected in the fluctuating validation curves.

- **Combined visible and infrared datasets:** Combining visible and infrared datasets introduces complementary features, but also unique challenges due to differences in feature distributions between the two modalities. These challenges, along with the relatively small size and limited diversity of each dataset, result in persistent validation instability. To address these issues, advanced fusion techniques and larger, more diverse datasets are required to improve generalization and model stability.

## CONCLUSION AND FUTURE DIRECTIONS

We presented an approach for FER by leveraging an attention-enhanced ResNet-18 model to recognize five expressions—angry, happy, neutral, sad, and surprised—from the VIRI database, and three expressions—fear, disgust, and happy—from the NVIE database. Our method includes both single-modal models for visible and infrared modalities and a multi-modal early fusion approach that combines pair-wise features from these modalities. By integrating an attention mechanism within this framework, our model demonstrates significant improvements in accuracy, precision, recall, and F1-score compared to previous methods. This attention mechanism enables our model to focus on salient features, especially in multi-modal fusion, leading to a more robust FER system.

While our results highlight the efficacy of the proposed method, several limitations warrant discussion. The primary limitation is the relatively small size of the VIRI and NVIE datasets, which may affect the model's ability to generalize to larger, more diverse populations. Although data augmentation was used to mitigate this limitation, issues of overfitting persisted, especially with the infrared modality, which is more sensitive to variations in ambient temperature and lighting. Furthermore, the model's performance, though strong in controlled environments, may decline in real-world settings where factors such as lighting, facial occlusions (*e.g.*, glasses, masks), and extreme facial angles are prevalent. Additionally, the reliance on specific facial features within each modality could mean that the model is less effective for expressions that are subtle or challenging to capture across both visible and infrared data.

Future research should address these limitations by focusing on larger, more diverse datasets to enhance model generalization and minimize overfitting. Using advanced data augmentation methods, such as generative adversarial networks (GANs) to generate synthetic data, could provide greater dataset diversity. Exploring transformer-based

architectures could offer significant advantages in FER, as their attention mechanisms may capture complex dependencies across modalities, potentially improving emotion recognition in diverse settings. Future directions should also investigate alternative fusion strategies to better integrate information from visible and infrared modalities, enhancing the model's ability to capture complementary features effectively. Furthermore, implementing ensemble-learning approaches, refining infrared data processing, and incorporating robust validation frameworks, including k-fold cross-validation, would strengthen model reliability and versatility.

### Funding

Adnan M. Abu-Mahfouz received funding by NextGen Enterprises and Institutions, Council for Scientific and Industrial Research, Pretoria, South Africa. The funders had no role in study design, data collection and analysis, decision to publish, or preparation of the manuscript.

### Grant Disclosures

The following grant information was disclosed by the authors:
NextGen Enterprises and Institutions, Council for Scientific and Industrial Research, Pretoria, South Africa.

### Competing Interests

The authors declare that they have no competing interests.

### Author Contributions

- Muhammad Tahir Naseem conceived and designed the experiments, performed the experiments, analyzed the data, performed the computation work, prepared figures and/or tables, and approved the final draft.
- Chan-Su Lee conceived and designed the experiments, analyzed the data, authored or reviewed drafts of the article, and approved the final draft.
- Tariq Shahzad performed the experiments, prepared figures and/or tables, and approved the final draft.
- Muhammad Adnan Khan conceived and designed the experiments, performed the computation work, prepared figures and/or tables, and approved the final draft.
- Adnan M. Abu-Mahfouz performed the experiments, analyzed the data, performed the computation work, authored or reviewed drafts of the article, funding Acquisition, and approved the final draft.
- Khmaies Ouahada performed the experiments, analyzed the data, authored or reviewed drafts of the article, and approved the final draft.

### Data Availability

    The raw data is available at GitHub and figshare:

- https://github.com/naseemmuhammadtahir/raw-data.
- Naseem, Muhammad Tahir (2025). Dataset.zip. figshare. Dataset. https://doi.org/10.6084/m9.figshare.28236605.v1.

The code and the Readme file are available in the Supplemental Files.

## Supplemental Information

Supplemental information for this article can be found online at http://dx.doi.org/10.7717/peerj-cs.2676#supplemental-information.

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
