# Peer review of "Facial expression recognition using visible and IR by early fusion of deep learning with attention mechanism"

_PeerJ Computer Science, doi:10.7717/peerj-cs.2676_

## Round 0.1 · original submission · Major Revisions

Dear Authors,

Thank you for submitting your article. Based on the reviewers' comments, your article has not yet been recommended for publication in its current form. However, we encourage you to address the concerns and criticisms of the reviewers and to resubmit your article once you have updated it accordingly. Please pay special attention to highlighting the real contribution, originality, and novelty. Additionally, the paper does not contain sufficient comparison with prior works. Furthermore, many of the figures have low resolution and should be polished.

Reviewer 3 has asked you to provide specific references. You are welcome to add them if you think they are relevant and useful. However, you are under no obligation to include them, and if you do not, it will not affect my decision.

Best wishes,


Reviewer 1 ·

Basic reporting

The manuscript follows the standard writing practices but introduction should contain more motivation for instance, add a flow diagram or architecture diagram to make the life easier for the readers. Also add a paragraph showing uniqueness of the work.

Experimental design

The details provided are detailed and in-depth.

Validity of the findings

Appropirate.

Reviewer 2 ·

Basic reporting

The paper is generally well-written, with no major grammatical errors, but there are areas where the language could be improved for clarity and fluency. Some phrases feel unnatural, such as "Measured was the intensity of the facial expression," which would be clearer as "The intensity of the facial expression was measured." Additionally, there is occasional inconsistency in verb tense usage, with the text switching between past and present tense in ways that disrupt the flow. Addressing these issues would make the paper easier to read.
The abstract, particularly its first part, is somewhat lengthy and could be more concise. It doesn’t immediately provide clear or essential information, so shortening this section would help capture the reader’s attention more effectively. The introduction provides a decent overview of the topic and offers enough context to understand the motivation behind the research. However, the hypothesis or objectives could be stated more explicitly to further clarify the purpose of the study.
In the Related Works section, the literature is well-referenced, and a wide range of relevant studies are cited. However, some statements come across as more opinion-based rather than grounded in empirical evidence. For example, the phrase "These methods might work well on some datasets, but when used in uncontrolled settings on more complicated expression datasets, their performance is probably going to suffer," lacks the backing of data or citations to support this claim. It would be beneficial to either provide evidence or references to substantiate such assertions or to rephrase them more cautiously.

Experimental design

The article falls within the Aims and Scope of the journal, and the code and methods are clearly presented, allowing for reproducibility. However, the novelty of the work is quite limited. The primary contribution of this paper is the addition of an attention layer to a pre-trained ResNet and the fusion of features from two ResNets using a simple concatenation approach. An ablation study is notably missing, which would be essential to assess whether the inclusion of the attention layer actually improves the model's accuracy compared to the same network without it.
Additionally, the authors cite a previous paper (ref. 10) where the first two authors are the same as in this work. In that paper, a similar method is presented, and they report exactly the same accuracy on the VIRI dataset as in this paper, which raises concerns about the novelty of the current work. I also recommend repeating the experiment using cross-validation to produce more reliable results.
When comparing the results with prior work, the article omits several relevant studies that use the same datasets. A simple search for the dataset names on Google Scholar yields several previous studies utilizing these datasets, which have not been cited. Properly referencing these works would provide a more comprehensive comparison of the proposed approach with the state-of-the-art methods.

Validity of the findings

The main issues with the paper are the lack of novelty in its contribution and the insufficient comparison with previous work.
The article lacks a meaningful discussion of the results, and the conclusions are quite weak. There is little to no assessment of the impact or novelty of the work, and while the paper presents a replicable approach, the rationale and benefit to the field are not clearly articulated. The experiments are presented, but without in-depth evaluation or analysis of their significance. This weakens the overall contribution of the paper and leaves important aspects of the study unexplored.

Additional comments

The article addresses a highly relevant and timely topic; however, in its current form, the paper should not be published due to its lack of novelty and insufficient comparison with prior work. The modifications—adding an attention layer to a pre-trained ResNet and using simple feature concatenation—are minimal, amounting to just a few lines of code, and do not constitute a significant contribution. Additionally, there is little discussion of the results, and the conclusions are weak. I suggest a more in-depth study comparing different CNNs, transformers, attention mechanisms, or feature fusion techniques to provide a more substantial contribution to the field.

Reviewer 3 ·

Basic reporting

Please find the attached file with all the comments on the review.

Experimental design

.

Validity of the findings

.

Additional comments

.

Annotated reviews are not available for download in order to protect the identity of reviewers who chose to remain anonymous.

---

## Round 0.2 · Minor Revisions

Dear Authors,

The concerns of Reviewer 2 are important and we encourage you to address these comments before resubmitting the manuscript.

Best wishes,

Reviewer 2 ·

Basic reporting

The article has improved after the revision, although its novelty remains somewhat limited.

Experimental design

no comment

Validity of the findings

The high instability in the validation curves is quite concerning. The good accuracy obtained seems to result from stopping the training at a moment when the value happens to peak, but stopping slightly earlier or later could lead to a drop of up to 20 points. This behavior might simply be due to the use of the CosineAnnealingLR scheduler, which may not pose a significant issue, or it could indicate poor model robustness, making the results potentially misleading. Therefore, I strongly suggest repeating the training without the scheduler to assess whether the instability is resolved. If the curves stabilize without the scheduler, it would confirm that the results are robust and valid. However, if the instability persists, it would point to a deeper issue with the model or its training process that needs to be addressed.

Reviewer 3 ·

Basic reporting

The authors addressed all my comments. I don't have any further comments on the paper.

Experimental design

The authors addressed all my comments. I don't have any further comments on the paper.

Validity of the findings

The authors addressed all my comments. I don't have any further comments on the paper.

Additional comments

The authors addressed all my comments. I don't have any further comments on the paper.

---

## Round 0.3 · Minor Revisions

Dear Authors,

According to Reviewer 2, your paper still needs a revision. We encourage you to address these minor comments before resubmitting the manuscript.

Best wishes,

Reviewer 2 ·

Basic reporting

no comments

Experimental design

no comments

Validity of the findings

The new data provided confirms the instability of the model, which limits the validity of the results. In this case, it cannot be claimed that the peak accuracy reflects the model's true accuracy. Therefore, either the model must be stabilized (by adjusting hyperparameters, reducing model complexity, etc.), or the reported results should present the average accuracy over a range of epochs.

Reviewer 3 ·

Basic reporting

no comment

Experimental design

no comment

Validity of the findings

no comment

Additional comments

The authors addressed all my comments. I don't have any further comments on the paper.

---

## Round 0.4 · accepted · Accept

Dear Authors,

Thank you for addressing the reviewers' comments. Your manuscript now seems ready for publication.

Best wishes,

Reviewer 2 ·

Basic reporting

No comments

Experimental design

No comments

Validity of the findings

No comments

Additional comments

No comments